# Development of a Model for Genomic Prediction of Multiple Traits in Common Bean Germplasm, Based on Population Structure

**DOI:** 10.3390/plants11101298

**Published:** 2022-05-12

**Authors:** Jing Shao, Yangfan Hao, Lanfen Wang, Yuxin Xie, Hongwei Zhang, Jiangping Bai, Jing Wu, Junjie Fu

**Affiliations:** 1Department of Crop Genetics and Breeding, College of Agronomy, Gansu Agricultural University, Lanzhou 730070, China; totoro_sj@163.com; 2Institute of Crop Sciences, Chinese Academy of Agricultural Sciences, Beijing 100081, China; haoyangfan2014@163.com (Y.H.); wanglanfen@caas.cn (L.W.); xieyuxin@caas.cn (Y.X.); zhanghongwei@caas.cn (H.Z.); 3Gansu Provincial Key Laboratory of Aridland Crop Science, Lanzhou 730070, China

**Keywords:** common bean germplasm, genomic prediction, population structure

## Abstract

Due to insufficient identification and in-depth investigation of existing common bean germplasm resources, it is difficult for breeders to utilize these valuable genetic resources. This situation limits the breeding and industrial development of the common bean (*Phaseolus vulgaris* L.) in China. Genomic prediction (GP) is a breeding method that uses whole-genome molecular markers to calculate the genomic estimated breeding value (GEBV) of candidate materials and select breeding materials. This study aimed to use genomic prediction to evaluate 15 traits in a collection of 628 common bean lines (including 484 landraces and 144 breeding lines) to determine a common bean GP model. The GP model constructed by landraces showed a moderate to high predictive ability (ranging from 0.59–0.88). Using all landraces as a training set, the predictive ability of the GP model for most traits was higher than that using the landraces from each of two subgene pools, respectively. Randomly selecting breeding lines as additional training sets together with landrace training sets to predict the remaining breeding lines resulted in a higher predictive ability based on principal components analysis. This study constructed a widely applicable GP model of the common bean based on the population structure, and encouraged the development of GP models to quickly aggregate excellent traits and accelerate utilization of germplasm resources.

## 1. Introduction

The common bean (*Phaseolus vulgaris* L.) is an edible legume with the widest and largest cultivation area in the world [1]. It is an important source of plant protein for humans and is high in protein, has a medium starch content, is low in fat, and is rich in mineral nutrients [2]. The generally low yield of the common bean limits its advantages; therefore, increasing the production will have an important impact on improving the nutrition and health of consumers, especially in developing countries [2]. Hence, genetic research and breeding improvement of bean germplasms are essential. The genetic resources of the common bean mainly consist of two highly differentiated gene pools: the Andean (An) and the Mesoamerican (M) [3,4]. These two gene pools contain abundant germplasm resources, including landraces and breeding lines.

During the domestication of the common bean, landraces on which humans depend for cultivation and diet are often grown in low-input production systems. They have a wide planting range and strong adaptability [5]. Alleles for endemic disease resistance and tolerance to major climatic stresses have also been reported [6]. A broad genetic basis for landraces has been identified in germplasm evaluation studies, and the diversity of breeding lines has narrowed through breeding and domestication [7,8]. Therefore, common bean landraces are a resource for increasing the genetic diversity of breeding lines. This suggests the possibility of predicting and identifying important agronomic traits and effective gene loci [9] using the genetic variation present in the existing common bean germplasm. A population with rich genetic diversity could serve as an effective training set in genetic assessment models based on molecular genetic markers.

Genomic prediction (GP) [10] has been widely used for food crops, such as barley [11], wheat [12], maize [13], and rice [14]. The construction of prediction models provides a good method for selecting promising varieties. It has three main advantages: it increases the genetic gain, shortens the breeding period, and reduces capital costs. In recent years, an increasing number of studies have focused on the GP of common bean. More advanced research mainly focuses on the breeding and development of the common bean, and the construction model of related single nucleotide polymorphisms (SNPs) is determined by combining genome-wide association analysis (GWAS) and GP [15,16,17,18]. Other studies have mainly focused on the genomic prediction of a few traits in common bean cultivars using different models [19,20,21]. Several studies have been conducted to improve model predictive ability by considering genotype-by-environment interactions (G×E) [22,23,24]. In particular, the common bean is a species with a strong population structure. In existing studies, population genetic structure has been considered in the GP of several types of crops with well-defined genetic population distribution and less domestication, such as wheat [25,26], rice [25], maize [27], and oats [28], to improve the predictive power. However, studies about modeling the distribution of common bean germplasm resources are limited.

In this study, we optimized the model’s precision to predict breeding lines by exploring the composition of the training set in the GP of the common bean (Figure 1). Fifteen traits, i.e., days to flowering (DF), days to maturity (DM), plant height (PH), stem node number (SNN), branch number (BN), pod per plant (PP), pod length (PL), pod width (PW), pod height (PDH), seeds per pod (SP), grain yield per plant (GY), 100-grain weight (GW), seed length (SL), seed width (SW), and seed height (SH), were used. The description of each trait was included in Appendix A [29]. First, prediction abilities were measured in two gene pools of common bean landraces using a ten-fold cross-validation repeated 100 times. Second, we analyzed the ability to combine or assign the landraces of the two gene pools to predict the breeding lines of both gene pools. Third, we considered the ability to add parts of breeding lines as a training set to predict the value of the remaining breeding lines. This study provides new perspectives for common bean breeding and offers a reference strategy for other crops with a strong population structure.

## 2. Results

### 2.1. Evaluation of Population Size and Marker Number on the Model’s Precision

Fifteen traits were used to evaluate the model’s predictive ability among 484 common bean landraces from the germplasm collection at the Chinese Academy of Agriculture Sciences (CAAS). We performed cross-validation using the ridge regression best linear unbiased prediction (RR-BLUP) model for landraces from two common bean gene pools, respectively. Multiple gradients of population size and number of SNPs were designed to examine the predictive ability of the combined landraces for multiple traits. This study found that the predictive ability of the landraces from the two gene pools differed greatly in different traits (0.35–0.82) (Appendix A, Appendix A), and that of the combined landraces was higher (0.59–0.88) (Figure 2, Appendix A). As shown in these figures, the predictive ability could be improved with an increase in the population size and number of SNPs, but it no longer showed significant improvement when it increased to a certain number. The optimal population size (opt_pop) and optimal number of SNPs (opt_snp) differed for each trait. In particular, the opt_pop of DF is 40% and opt_snp is 2000 (0.67), the opt_pop of PH is 40% and opt_snp is 1000 (0.84), the opt_pop of SP is 40% and opt_snp is 500 (0.77), and the opt_pop of SH is 40% and opt_snp is 500 (0.81) (Appendix A). However, the predictive ability of each trait ranged from 0.56 for GY to 0.85 for GW at the opt_pop and opt_snp. These results illustrated that landraces as a training set could obtain better predictive ability for all investigated traits, which lays the foundation for population selection to optimize the prediction of breeding lines.

### 2.2. Ability of Landrace Subgene Pools in Predicting Breeding Lines

We designed a pattern of landrace predicting breeding lines based on two gene pools (An and M) of the common bean. The ability of landraces to predict breeding lines ranged from 0.54 to 0.82, except for two traits that were more affected by photoperiod, DF, and DM (0.32 and 0.21, respectively) (Table 1). A few differences were found when comparing the ability of landraces (An + M) and landraces (An) to predict breeding lines (An). This ability was also comparable when landraces (An + M) and landraces (M) were used to predict breeding lines (M; Appendix A). Subsequently, we counted the accessions corresponding with the predicted and observed values from the top 30 and top 30% for each trait. The number of the same accessions was recorded as the ratio (predicted/observed) of the top 30 and the top 30% of the number of materials. The results showed that the ratios differed for each trait. However, there was no significant difference in the ratios of predicted breeding lines (An) using landrace (An + M), or predicted breeding lines (An) using landrace (An) for the same trait. Similar results were found for the Mesoamerican gene pools (Appendix A). Since combining and assigning gene pools had no significant effect on the landraces’ ability to predict breeding lines, we do not recommend assigning landrace pools as the training set in the subsequent prediction studies.

### 2.3. Optimization of Landrace Training Sets by Adding Breeding Lines

The genetic data of the common bean were transformed using principal components analysis (PCA) into components explaining most of the genetic variations. The first three axes of PCA explained 63.23% of the variability in the entire collection. The two gene pools of the common bean, Andean and Mesoamerican, were undoubtedly distinguished as two clusters (Figure 3). Based on the common bean germplasm, we designed a two-step scenario to optimize the composition of the training set by adding 50% of the breeding lines (Table 2). Firstly, **scenario 1** investigated the impact of adding breeding lines as the training set on the predictive ability of the two gene pools. In the prediction using 50% Andean gene pool breeding lines (A2) as the testing set, the predictive ability of the model was improved in all traits to varying degrees compared to when landraces and 50% breeding lines (A1 or A1 + M1) were used as the training set. Adding 50% breeding lines (M1 or A1 + M1) to the prediction using 50% Mesoamerican gene pool breeding lines (M2) as the testing set also improved the model’s precision. However, the addition of breeding lines was more stable for improving the Andean model’s precision (Appendix A). Then, **scenario 2** made a prediction comparison on how to select 50% of the individuals in the breeding lines. The results showed that compared with the 50% breeding lines randomly selected as the additional training set, the 50% breeding lines (A1 + M1) extracted from the subgene pool as the additional training set had a more stable and improved predictive ability of the model, especially for the traits PL (+0.12) and PW (+0.25) (Table 3).

## 3. Discussion

Landraces are accessions with rich genetic diversity and are widely used in the genomic prediction of crop germplasms. Moderate to high prediction abilities were obtained for accessions such as wheat [30,31], maize [32], and white lupin [33]. Our results are in concordance with those of previous GP studies on landraces. The model’s precision ranged from 0.59 for GY to 0.88 for PH. We evaluated the model’s precision when the landraces were used as training sets and breeding lines were used as testing sets. The GP model built by combining the gene pool landraces (An + M) did not significantly improve the ability to predict breeding lines (An) or breeding lines (M). We speculated that the main reason could be that the common bean is domesticated from wild relatives, which are characterized by their inhabitation in a relatively narrow ecological niche [34]. Andean and Mesoamerican gene pools are probably two independent origin centers of the common bean before domestication, which resulted in the inability to communicate genetic information between the two gene pools [35]. Therefore, the model’s precision was not substantially improved when the training set consisted of landraces (An + M) that combined the two gene pools. Subsequently, the kinship between the training set (landraces) and the testing set (breeding lines) was examined using PCA (Figure 3). When a part of the breeding lines was added to the training set as an additional training set, the model’s precision was improved across multiple traits. It was observed that the breeding lines selected based on the gene pool resulted in more stable and more accurate improvements of the model than the randomly selected breeding lines.

The genetic relationship between the training sets and testing sets is crucial when using GP to select promising breeding materials from a germplasm collection. Understanding the population structure of germplasm resources and ensuring the distribution of genetically related individuals in the training sets and testing sets could solve the problem of genetic relationships in structured populations and maximize predictive ability [26,27,36]. In this study, **scenario 1** indicated that we artificially increased the relevance between the training sets and the testing sets by adding a part of the breeding lines as the additional training sets. The GP model predicted the remaining breeding lines more accurately. In the Andean gene pool, compared to G1, the prediction abilities of G2 and G3 were improved, especially in GW (+0.27, +0.27) and SW (+0.23, +0.24). In the Mesoamerican gene pool, the prediction abilities of G5 and G6 were improved for traits such as DF (+0.18, +0.17) and SNN (+0.14, +0.14) compared to that of G4 (Table 3). The PCA indicated that accessions within the Andean gene pool were more highly related to each other than accessions within the Mesoamerican gene pool (Figure 3). The prediction abilities for different traits in the two gene pools were related to the genetic correlation between the germplasm in each subgene pool. Similarly, in a genomic prediction study of four traits (heading, height, biomass, and yield) in oats, it has been found that the multi-group training sets will obtain higher predictive ability (0.32–0.87) than across-group scenarios (−0.55–0.27) [28].

The population structure plays a critical role in optimizing the composition of the training set. In a rice population with a strong structure, stratified sampling has higher accuracy than other random sampling methods for four traits: florets per panicle, flowering time, plant height, and protein content [25]. The stratified sampling here was based on several clusters of the rice population structure in PCA. This was the same as the sub-gene pool sampling of the common bean population in **scenario 2** in this study. For all traits, adding extracted breeding lines from the subgene pool as an additional training set is beneficial for improving the ability to predict the remaining breeding lines. The predictive ability of G7 and G8 showed a remarkable increase for two traits, PL and PW, compared to that of other traits. Our findings are consistent with those of previous studies of GP in similarly structured populations. For example, in spring wheat germplasm, Muleta et al. [26] purposefully divided accessions into two major subpopulations through population structure analysis. On this basis, the training sets were selected to predict the testing sets, and promising results (≥0.7) were obtained for disease resistance traits such as infection type and disease severity. In the oat population, the prediction abilities of the testing sets with different training set organizations showed significant differences. For oat yield [28], selection based on population structure could increase the predictive ability of the training set to the testing set when compared with random selection. Yield prediction was improved by 0.35 and 0.03 in two predicted subpopulations of oats.

However, there is evidence that considering the population structure to optimize the training set can improve the model’s precision [37,38,39]. For the first time, the optimized training set was used in two major gene pools to predict the breeding lines in common bean germplasm resources. This study provides new ideas for utilizing common bean germplasm resources in China and a new breeding strategy for similar small crops in the primary domestication stage. The effective use of the collected genetic diversity data provided help for further breeding and research on the common bean.

## 4. Materials and Methods

### 4.1. Plant Materials

The common bean dataset used in this study, i.e., the Chinese Academy of Agriculture Sciences (CAAS) collection determined and evaluated by Wu et al. [29], comprised 628 germplasm resources. This included 484 landraces (including 223 from the Andean gene pool and 261 from the Mesoamerican gene pool) and 144 breeding lines (including 60 from the Andean gene pool and 84 from the Mesoamerican gene pool).

### 4.2. Genotyping

Genotyping data for 4.8 million SNPs distributed over all 11 chromosomes published in an article by Wu et al. [29] were used to investigate the model’s precision for 628 common bean landraces and breeding lines. In this study, we have processed these data for genomic prediction analysis. After filtering on the minor allele frequency (>5%) and the percentage of missing values (<50%) using VCFtools version 0.1.13 (https://github.com/vcftools/vcftools, accessed 2 March 2021), the sample function in R version 4.0.2 (https://www.r-project.org/) was used to sample 11 chromosomes according to the proportion of SNPs, and 9781 SNP markers were obtained.

### 4.3. Phenotyping

Phenotypic data for 15 traits in the common bean dataset from the published article by Wu et al. [29] were analyzed: days to flowering (DF), days to maturity (DM), plant height (PH), stem node number (SNN), branch number (BN), pod per plant (PP), pod length (PL), pod width (PW), pod height (PDH), seeds per pod (SP), grain yield per plant (GY), 100-grain weight (GW), seed length (SL), seed width (SW), and seed height (SH). The experiments were carried out from 2014 to 2016 at three locations (i.e., Bijie, Harbin, and Sanya in China) [29]. Phenotypic data for each trait were calculated from the average of nine location–year combinations.

### 4.4. Genomic Prediction Model and Analysis

The genomic estimated breeding values (GEBVs) were calculated using the ridge regression best linear unbiased prediction (RR-BLUP) of the GP model. The model assumes that the effects of all SNPs have a common variance, and a stable predictive ability could be obtained in most traits (especially traits influenced by a large number of minor genes), which is implemented in the R package “*rrBLUP*”. The *rrBLUP* package [40] can solve any mixed model of the form:y=Xβ+Zu+ε
where y is an *N* × 1 vector of observed phenotypes (*N* is the size of the training set), X is the identity matrix for the fixed effects β, Z is an *N* × *M* genotypic matrix for the random effects u with $u\sim N\left(0,I\sigma_{u}^{2}\right)$ (*M* is the number of SNPs), and ε is the vector of residuals with $\varepsilon \sim N\left(0,I\sigma_{u}^{2}\right)$, where I is the identity matrix. In the Z matrix, {0,1,2} for biallelic SNP markers of genotypes (i.e., 0 and 2 for homozygotes and 1 for heterozygotes). The common bean is a highly self-pollinated crop [2] and its genotype data are composed of {0,1}.

In this study, we conducted three GP analyses based on this model. (1) Seven different population sizes (10, 20, 40, 50, 60, 80, and 100%) and eleven different numbers of SNPs (100, 500, 1000, 1500, 2000, 3000, 4000, 5000, 6000, 7000, and 9781) were used in a ten-fold cross-validation with 100 replications for 15 agronomic traits in 484 common bean landraces. (2) Combining gene pool prediction: landraces (An + M) were used as the training set to predict breeding lines (An) and breeding lines (M). Assigning gene pool prediction: landraces (An) were used as the training set to predict breeding lines (An), and landraces (M) were used as the training set to predict breeding lines (M). (3) Two scenarios were designed: adding 50% breeding lines in different gene pools as additional training sets to predict the remaining 50% of breeding lines in each gene pool, selecting by gene pool; or randomly selecting the added breeding lines to predict the remaining individuals. The predictive ability was calculated using the mean of Pearson’s correlation coefficient (*r*) between the estimated and observed values in the testing set [41]. The closer the *r*-value is to 1, the higher the model’s predictive ability. The figures and tables in the article were generated using the R package *ggplot2* (https://cran.r-project.org/package=ggplot2, accessed on 2 March 2021) and Microsoft Excel 2019.

## 5. Conclusions

This study used existing common bean germplasm resources to perform genomic prediction in multiple traits. The research showed the following. (1) The number of markers and the size of the training population required to build a GP model for different traits of the common bean differ according to their heritability levels. For traits with high heritability, the number of markers required by the GP model and the size of the training population was smaller than those with low heritability. (2) Using the common bean as an example, the construction of the GP model for germplasm resource prediction using landraces and the optimization of the training population were analyzed. The main result was that adding a part of the breeding lines as an additional training set could improve the predictive ability of the GP model built by landraces on the remaining breeding lines. By analyzing the population structure of all accessions, the training population was optimized and the traits were predicted more accurately. This investigation provides an argument for mining and analyzing germplasm resources through GP for crops with a population structure similar to that of the common bean and provides a reference for germplasm selection for breeders.

## Figures and Tables

**Figure 1 plants-11-01298-f001:**
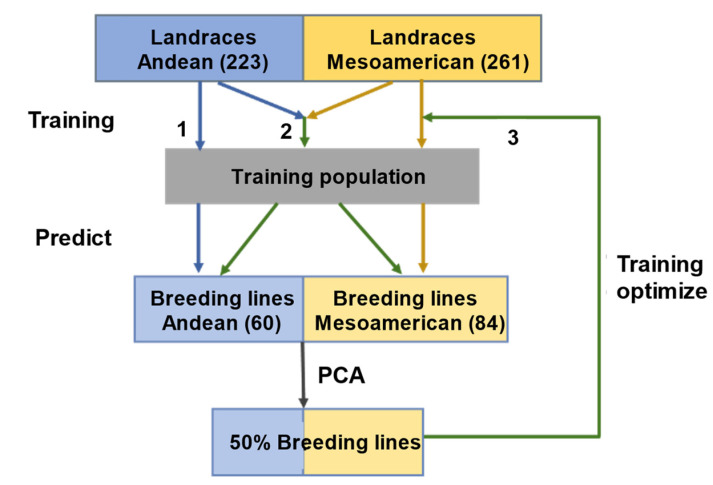
A workflow for genomic prediction of common bean germplasm. The above three ways are represented by the flow of the three numbers (1, 2, 3) in the figure. (1) The landraces from each of two subgene pools were used as training sets to predict breeding lines, respectively. (2) All landraces were used as training sets to predict breeding lines. (3) A part of the breeding lines was selected and included in the training sets.

**Figure 2 plants-11-01298-f002:**
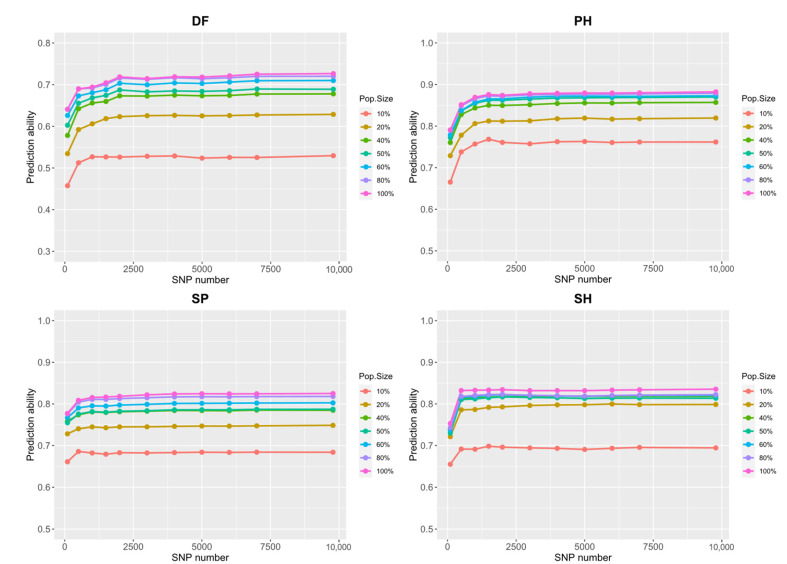
Different number of SNPs and different population sizes were used in ten-fold cross-validation using 100 replications for 15 agronomic traits in 484 common bean landraces (The figure shows four traits: DF, days to flowering; PH, plant height; SP, seeds per pod; and SH, seed height. Other traits are shown in Appendix A).

**Figure 3 plants-11-01298-f003:**
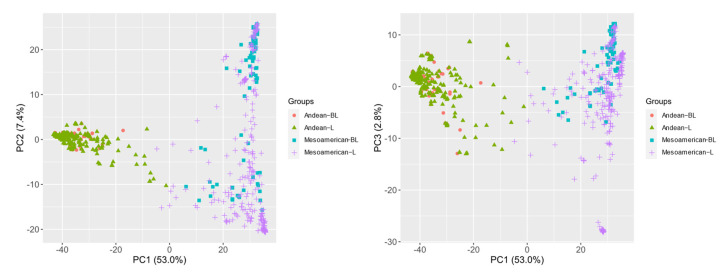
The three principal components of a PCA performed on 9781 SNPs markers among 628 common bean accessions. The four colors (shapes) represent landraces and breeding lines in the two gene pools.

**Table 1 plants-11-01298-t001:** The predictive ability of landraces as the training set to predict breeding lines using 15 traits.

Traits	Landraces (An + M)			Landraces (An)	Landraces (M)
	Breeding Line (An + M)	Breeding Line (An)	Breeding Line (M)	Breeding Line (An)	Breeding Line (M)
DF	0.3204	0.1699	0.2063	0.1832	0.2043
DM	0.2115	0.3917	−0.0045	0.4370	−0.1070
PH	0.6761	0.4201	0.6536	0.4035	0.6257
SNN	0.6492	0.3492	0.3865	0.2747	0.2640
BN	0.5365	0.4573	0.5907	0.4414	0.5954
PP	0.7797	0.1974	0.6856	0.2731	0.6967
PL	0.7646	0.5688	0.8426	0.5786	0.8441
PW	0.6702	0.5927	0.6848	0.5574	0.6374
PDH	0.7267	0.4896	0.5255	0.5088	0.5085
SP	0.7818	0.2422	0.4995	0.2645	0.3562
GY	0.5931	0.2096	0.4799	0.1810	0.4886
GW	0.7616	0.3387	0.6106	0.3325	0.6046
SL	0.7705	0.4458	0.7070	0.4744	0.7222
SW	0.7195	0.4672	0.6254	0.4007	0.6146
SH	0.8155	0.7401	0.6924	0.7330	0.6917

DF, days to flowering; DM, days to maturity; PH, plant height; SNN, stem node number; BN, branch number; PP, pod per plant; PL, pod length; PW, pod width; PDH, pod height; SP, seeds per pod; GY, grain yield per plant; GW, 100-grain weight; SL, seed length; SW, seed width; SH, seed height.

**Table 2 plants-11-01298-t002:** Two-step scenario for optimizing the training set by adding 50% of the breeding lines.

Scenarios	Scenario 1						Scenario 2	
Groups	G1	G2	G3	G4	G5	G6	G7	G8
Training sets	Landraces An + M(484)	Landraces An + M(484)	Landraces An + M(484)	Landraces An + M(484)	Landraces An + M(484)	Landraces An + M(484)	Landraces An + M(484)	Landraces An + M(484)
Additional training sets		Breeding lines A1 (30)	Breeding lines A1 + M1 (72)		Breeding lines M1 (42)	Breeding lines A1 + M1 (72)	Breeding lines A1 + M1 (72)	Breeding lines random (72)
Testing sets	Breeding lines A2 (30)	Breeding lines A2 (30)	Breeding lines A2 (30)	Breeding lines M2 (42)	Breeding lines M2 (42)	Breeding lines M2 (42)	Breeding lines A2 + M2 (72)	Breeding lines remaining (72)

The 50% breeding lines (An) were randomly selected as breeding lines A1 (30). The remaining breeding lines (An) are used as breeding lines A2 (30). The 50% breeding lines (M) were randomly selected as breeding lines M1 (42). The remaining breeding lines (M) are used as breeding lines M2 (42). The 50% breeding lines were randomly selected as breeding lines random (72). The remaining breeding lines are used as breeding lines remaining (72).

**Table 3 plants-11-01298-t003:** The model’s precision for 8 groups across two scenarios among 15 traits.

Scenarios	Scenario 1						Scenario 2	
Groups	G1	G2	G3	G4	G5	G6	G7	G8
DF	0.2566	0.3590	0.3625	0.2109	0.3934	0.3800	0.4664	0.3686
DM	0.5108	0.6442	0.6237	−0.0846	0.0547	0.0323	0.3868	0.4308
PH	0.4958	0.6294	0.6446	0.6463	0.7818	0.7795	0.7904	0.8260
SNN	0.3920	0.5399	0.5549	0.4497	0.5869	0.5865	0.7255	0.6850
BN	0.5538	0.6950	0.6820	0.6053	0.6921	0.6996	0.6779	0.6922
PP	0.2465	0.2771	0.2675	0.7380	0.7356	0.7398	0.7986	0.7385
PL	0.6740	0.8103	0.8111	0.8934	0.9390	0.9456	0.8929	0.7724
PW	0.6559	0.7840	0.7854	0.7662	0.8975	0.8966	0.8874	0.6407
PDH	0.3626	0.3772	0.3631	0.3312	0.3355	0.3316	0.6837	0.5964
SP	0.1611	0.3168	0.3504	0.5389	0.5855	0.5877	0.7864	0.7056
GY	0.2468	0.2496	0.2515	0.5795	0.6191	0.6210	0.6440	0.5802
GW	0.2613	0.5268	0.5274	0.6238	0.6165	0.6155	0.7804	0.6944
SL	0.3876	0.4674	0.4775	0.7364	0.7285	0.7226	0.7647	0.6956
SW	0.4314	0.6614	0.6680	0.7061	0.7596	0.7624	0.8137	0.7460
SH	0.7114	0.7988	0.7988	0.6650	0.7389	0.7290	0.8535	0.8186

DF, days to flowering; DM, days to maturity; PH, plant height; SNN, stem node number; BN, branch number; PP, pod per plant; PL, pod length; PW, pod width; PDH, pod height; SP, seeds per pod; GY, grain yield per plant; GW, 100-grain weight; SL, seed length; SW, seed width; SH, seed height.

## Data Availability

Not applicable.

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
