# Peer review of "Development of a Model for Genomic Prediction of Multiple Traits in Common Bean Germplasm, Based on Population Structure"

_plants, 2022, doi:10.3390/plants11101298_

Round 1

Reviewer 1 Report

Generally, common bean germplasm has a huge importance as a nutritional source. The paper 'Beans (Phaseolus spp.) – model food legumes' by W. Broughton, Plant and Soil 252: 55–128, 2003 might be an interesting source to improve this paper.

In this version, the manuscript has not a 'reading flow'; many weak spots.

Chapter 'Results' not indicated; the clarity of the content is not given; also the relevance of the selected  phenotyping traits (for example in the 'Introduction' part;  

Comments/remarks: see file

Author Response

Thank you very much for your comments on revisions to the manuscript. We have answered and revised each question carefully. Please see the attachment for details.

Reviewer 2 Report

The manuscript describes an interesting and useful study that can make a significant contribution to common bean breeding. However, the structure of the manuscript and the way the study is described reduce its quality and lack of comprehension. Therefore, in order to accept the manuscript, it would be desirable to make improvements to it:
- Language needs to be significantly improved: in some places, it is difficult to understand the nature of the information provided.
- Words are often repeated in one sentence (including in the title), some of which I marked in the attached file, but they should be reviewed throughout the text.
- I would suggest changing the title of the manuscript, both to exclude repeats and to make it simpler and easier to understand.
- The use of terms needs to be considered, in many places several names are used at the same term at the same time. I marked part of the attached file, but I should check the whole text.

Some specific comments:

- Introduction - The manuscript does not separate Introduction from Results (see comment in attached file).

- Materials and Methods:

  • Plant material - It is not clear from the description whether the data were obtained during this study or from a previous study. This should be clarified.
  • When describing genotyping, reference should be made to publication or a description of the SNP methodology used. The current description contains more data processing methodologies than the use of markers.
  • Also in the case of phenotyping data, it is not clear whether the data were collected during this study or used from a previous study. This does not detract from the scientific value of the manuscript, but it must be clarified.

Author Response

(The authors gave the same response as above.)

Round 2

Reviewer 1 Report

The revision improved this manuscript markedly. It has now reached the state of acceptance for publication.

Author Response

Thank you very much. We greatly appreciate your efforts to improve our manuscript. We have carefully revised the language and spelling details. Please refer to the new manuscript.